# LARGE LANGUAGE MODELS ARE EFFECTIVE TEXT RANKERS WITH PAIRWISE RANKING PROMPTING

## ABSTRACT

Ranking documents using Large Language Models (LLMs) by directly feeding the query and candidate documents into the prompt is an interesting and practical problem. However, there has been limited success so far, as researchers have found it difficult to outperform fine-tuned baseline rankers on benchmark datasets. We analyze pointwise and listwise ranking prompts used by existing methods and argue that off-the-shelf LLMs do not fully understand these challenging ranking formulations. In this paper, we propose to significantly reduce the burden on LLMs by using a new technique called *Pairwise Ranking Prompting* (PRP). Our results are the first in the literature to achieve state-of-the-art ranking performance on standard benchmarks using moderate-sized open-sourced LLMs. On TREC-DL2020, PRP based on the Flan-UL2 model with 20B parameters outperforms the previous best approach in the literature, which is based on the blackbox commercial GPT-4 that has 50x (estimated) model size, by over 5% at NDCG@1. On TREC-DL2019, PRP performs favorably with supervised models and is only inferior to the GPT-4 solution among LLM-based methods on the NDCG@5 and NDCG@10 metrics, while outperforming other LLM-based solutions, such as InstructGPT which has 175B parameters, by over 10% for all ranking metrics. By using the same prompt template on seven BEIR tasks, PRP beats supervised baselines and outperforms the blackbox commercial ChatGPT solution by 4.2% and pointwise LLM-based solutions by over 10% on average NDCG@10. Furthermore, we propose several variants of PRP to improve efficiency and show that it is possible to achieve competitive results even with linear complexity. We also discuss other benefits of PRP, such as supporting both generation and scoring LLM APIs, as well as being insensitive to input ordering.

## 1 INTRODUCTION

Large Language Model (LLMs) such as GPT-3 (Brown et al., 2020) and PaLM (Chowdhery et al., 2022) have demonstrated impressive performance on a wide range of natural language tasks, achieving comparable or better performance when compared with their supervised counterparts that are potentially trained with millions of labeled examples, even in the zero-shot setting (Kojima et al., 2022; Agrawal et al., 2022; Huang et al., 2022; Hou et al., 2023).

However, there is limited success for the important text ranking problem using LLMs (Ma et al., 2023). Existing results usually significantly underperform well-trained baseline rankers (e.g., Nogueira et al. (2020); Zhuang et al. (2023)). The only exception is a recent approach proposed by Sun et al. (2023), which depends on the blackbox, giant, and commercial GPT-4 system. Besides the technical concerns such as sensitivity to input order (ranking metrics can drop by more than 50% when the input document order changes), we argue that relying on such blackbox systems is not ideal for academic researchers due to significant cost constraints and access limitations to these systems, though we do acknowledge the value of such explorations in showing the capacity of LLMs for ranking tasks.

In this work, we first discuss why it is difficult for LLMs to perform ranking tasks with existing methods, specifically, the pointwise and listwise formulations. For pointwise approaches, ranking requires LLMs to output *calibrated* prediction probabilities before sorting, which is known to be very difficult and is not supported by the generation-only LLM APIs (such as GPT-4). For listwise approaches, even with instructions that look very clear to humans, LLMs can frequently generate

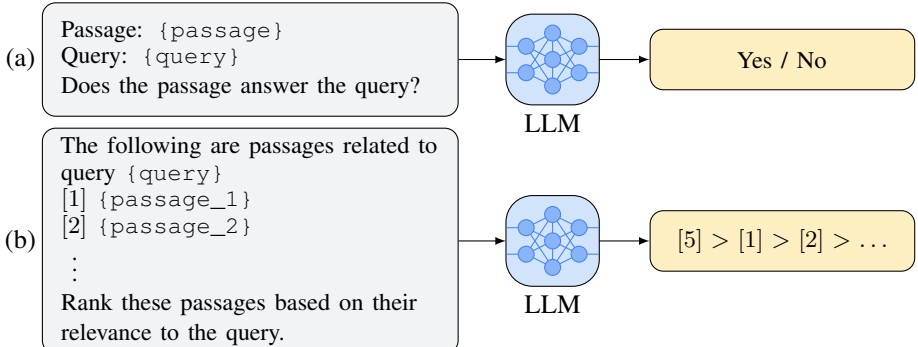

Figure 1: Two existing prompting methods for ranking: (a) the pointwise relevance generation approach and (b) the listwise permutation approach.

conflicting or useless outputs. Empirically we find that listwise ranking prompts from existing work generate completely useless outputs on moderate-sized LLMs. Such observations show that existing popular LLMs do not fully understand ranking tasks, potentially due to the lack of ranking awareness during their pre-training and (instruction) fine-tuning procedures.

We propose the Pairwise Ranking Prompting (PRP) paradigm, which uses the query and a pair of documents in the prompt for LLMs to perform ranking tasks, with the motivation to significantly reduce the task complexity for LLMs and resolve the calibration issue. PRP is based on simple prompt design and naturally supports both generation and scoring LLMs APIs. We describe several variants of PRP to address efficiency concerns. PRP results are the first in the literature that can achieve state-of-the-art ranking performance by using *moderate-sized, open-sourced* LLMs on standard benchmark datasets. On TREC-DL2020, PRP based on the FLAN-UL2 model with 20B parameters outperforms the previous best approach in the literature, based on the blackbox commercial GPT-4 that has (an estimated) 50X model size, by over 5% at NDCG@1. On TREC-DL2019, PRP is only inferior to the GPT-4 solution on the NDCG@5 and NDCG@10 metrics, but can outperform existing solutions, such as InstructGPT which has 175B parameters, by over 10% for nearly all ranking metrics. We also show competitive results using FLAN-T5 models with 3B and 13B parameters, demonstrating the power and generality of PRP. The observations are further validated on seven BEIR datasets covering various domains, where PRP performs competitively with supervised rankers and outperforms other LLM based approaches by a large margin. We further discuss other benefits of PRP, such as being insensitive to input ordering.

In summary, the contributions of this paper are three-fold:

- We, for the first time in published literature, show pairwise ranking prompting effectiveness for ranking with LLMs. It is able to produce state-of-the-art ranking performance on a wide range of datasets with simple prompting and scoring mechanism.

- Our results are based on moderate-sized, open-sourced LLMs, comparing with existing solutions that use blackbox, commercial, and much larger models. The finding will facilitate future research in this direction.

- We study several efficiency improvements and show promising empirical performance.

## 2 DIFFICULTIES OF RANKING TASKS FOR LLMS

As discussed in Section 1, to date there is limited evidence showing LLM-based rankers can outperform fine-tuned ones. We discuss why this is the case by overviewing and analyzing existing methods, which can be categorized into pointwise or listwise approaches.

### 2.1 POINTWISE APPROACHES

Pointwise approaches are the major methods prior to very recent listwise approaches discussed in Section 2.2. There are two popular methods, relevance generation (Liang et al., 2022) and query

generation (Sachan et al., 2022). Figure 1 (a) shows the prompt used for relevance generation. The relevance score $s_i$ is defined as:

$$s_i = \begin{cases} 1 + p(\text{Yes}), \text{if output Yes} \\ 1 - p(\text{No}), \text{if output No} \end{cases} \quad (1)$$

where $p(\text{Yes})$ and $p(\text{No})$ denote the probabilities of LLMs generating 'Yes' and 'No' respectively. Meanwhile query generation approach asks LLMs to generate a query based on the document ("Please write a question based on this passage. Passage: {{passage}} Question:"), and measures the probability of generating the actual query. Readers can refer to Sachan et al. (2022) for more details.

There are two major issues with pointwise approaches. First, pointwise relevance prediction requires the model to output *calibrated* pointwise predictions so that they can be used for comparisons in sorting. This is not only very difficult to achieve across prompts (Desai & Durrett, 2020), but also unnecessary for ranking, which only requires *relative* ordering. In fact, a major focus of the learning to rank field (Liu, 2009) in the information retrieval field is based on this observation. Also, pointwise methods will not work for generation API, which is common, such as GPT-4, since it requires the log probability of the desired predictions to perform sorting.

## 2.2 LISTWISE APPROACHES

Very recently, two parallel works (Sun et al., 2023; Ma et al., 2023) explore listwise approaches, by directly inserting the query and a list of documents into a prompt. Both methods feed a partial list of 10 or 20 documents every time and perform a sliding window approach due to the prompt length constraints. Figure 1 (b) shows a simplified version of the listwise ranking prompt. Both works explored text-davinci-003, i.e., InstructGPT (Ouyang et al., 2022) with 175B parameters, showing significantly worse performance than fine-tuned baseline rankers. Sun et al. (2023) were able to further explore gpt-3.5-turbo (the model behind ChatGPT) and GPT-4. Only the GPT-4 based approach could achieve competitive results, which is based on the blackbox, commercial, and giant (1T estimated parameters (VanBuskirk, 2023; Baktash & Dawodi, 2023)) system, without academic publication discussing technical details (OpenAI (2023) mainly focused on evaluations).

The issues are again due to the difficulty of the listwise ranking task for LLMs. Sun et al. (2023) show that there are frequent prediction failures with the following patterns:

- Missing: When LLMs only outputs a partial list of the input documents.
- Rejection: LLMs refuse to perform the ranking task and produce irrelevant outputs.
- Repetition: LLMs output the same document more than once.
- Inconsistency: The same list of documents have different output rankings when they are fed in with different order or context.

In fact, we tried the same prompt from (Sun et al., 2023) on the FLAN-UL2 model with 20B parameters, and found very few of the outputs to be usable. The model will either just output few documents (e.g., "[1]"), an ordered list based on id (e.g. "[3] > [2] > [1] ..."), or text which is not parseable.

Different from pointwise approaches, listwise approaches can only use the generation API – getting the log probability of all listwise permutations is prohibitively expensive. In other words, there is no good solution if the generation API does not output desired results, which is common. These methods will fall back to the initial ranking, and due to the high failure rate, the results are highly sensitive to input ordering.

These observations are not entirely surprising. Existing popular LLMs are generally not specifically pre-trained or fine-tuned against ranking tasks. However, we next show that LLMs do have a sense of pairwise relative comparisons, which is much simpler than requiring a calibrated pointwise relevance estimation or outputting a permutation for a list of documents.

## 3 PAIRWISE RANKING PROMPTING

We propose Pairwise Ranking Prompting (PRP) for ranking with LLMs. We describe the basic pairwise prompting unit, how it supports both generation and scoring APIs, and propose several variants of PRP with different ranking strategies and efficiency properties.

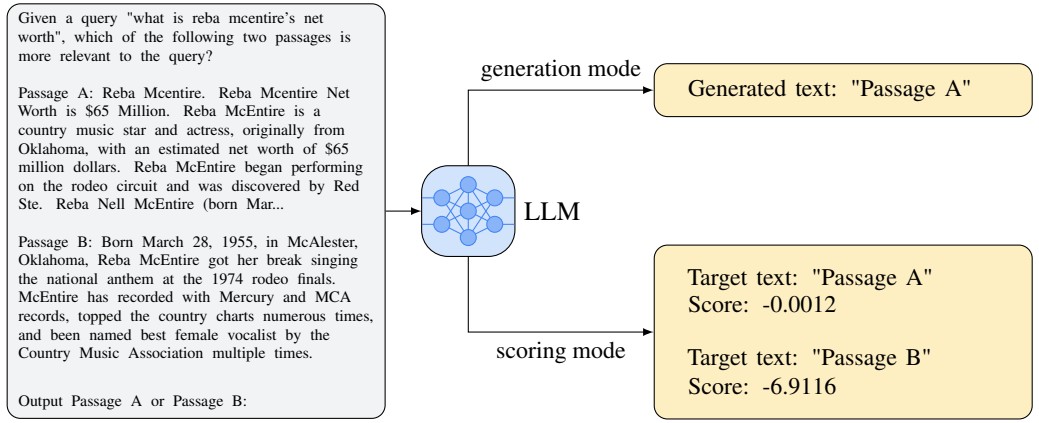

Figure 2: An illustration of pairwise ranking prompting. The scores in scoring mode represent the log-likelihood of the model generating the target text given the prompt.

## 3.1 PROMPTING DESIGN

Our pairwise ranking prompt is simple and intuitive, as shown in Figure 2. The exact prompt template is shown in Appendix F. This pairwise prompting will serve the basic computation unit in all PRP variants, which we denote as $u(q, d_1, d_2)$ for a query $q$ and two documents $d_1$ and $d_2$.

PRP naturally supports both generation API and scoring API. The latter is made possible since we only have two expected outputs ("Passage A" and "Passage B") for LLM inquiries. Since using scoring mode can mitigate potential issues when the generation API generates irrelevant outputs, our main results are based on the scoring mode, though we show there are very few prediction failures and provide comparisons between these two modes in Appendix B.

Since it is known that LLMs can be sensitive to text orders in the prompt (Lu et al., 2022; Liu et al., 2023a), for each pair of documents, we will inquire the LLM twice by swapping their order: $u(q, d_1, d_2)$ and $u(q, d_2, d_1)$.

The output of the pairwise ranking prompting is a local ordering of $d_1 > d_2$ or $d_2 > d_1$ if both promptings make consistent decisions, and $d_1 = d_2$ otherwise. Next we discuss three variants of PRP using the output of pairwise ranking prompting as the computation unit. We note that pairwise comparison can serve as the basic computation unit of many algorithms (e.g., selection algorithm) and leave other alternatives for future work.

## 3.2 ALL PAIR COMPARISONS

We enumerate all pairs and perform a global aggregation to generate a score $s_i$ for each document $d_i$. We call this approach PRP-Allpair. Specifically, we have:

$$s_i = 1 \cdot \sum_{j \neq i} \mathbb{I}_{d_i > d_j} + 0.5 \cdot \sum_{j \neq i} \mathbb{I}_{d_i = d_j}. \tag{2}$$

Intuitively, if the LLM consistently prefers $d_i$ over another document $d_j$, $d_i$ gets one point. When LLM is not sure by producing conflicting or irrelevant results (for the generation API), each document gets half a point. There might be ties for the aggregated scores, in which case we fall back to initial ranking. In this work, we use equation 2 which works for both scoring and generation APIs, and note there could be other ways to weight the scoring function, such as leveraging prediction probabilities in scoring mode.

PRP-Allpair favors simple implementation (all LLM API calls can be executed in parallel), and is highly insensitive to input ordering. The clear drawback is its costly $O(N^2)$ calls to LLM APIs, where $N$ is the number of documents to be ranked for each query.

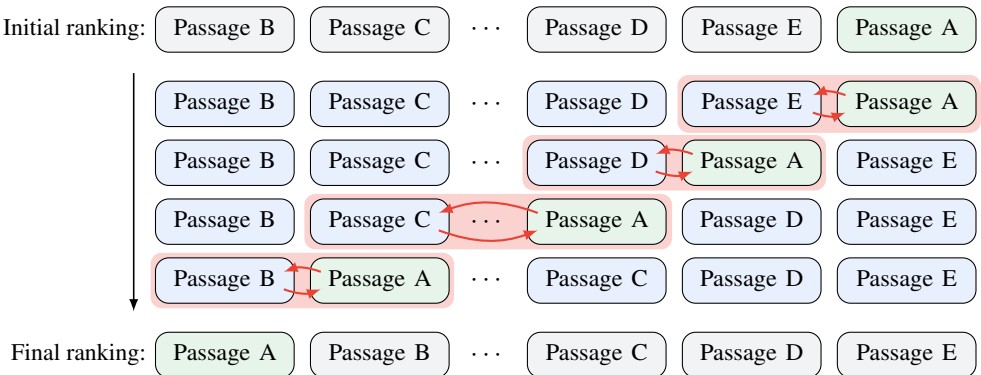

Figure 3: An illustration of one pass of our sliding window approach. Starting from right to left, we compare each document pair and swap it if the LLM output disagrees with the initial ranking. We can see that the sliding window approach is able to bring up initially lower ranked "Passage A" (shown in green) to the top of the ranking. $K$ such passes will ensure a high-performing top-$K$ ranking.

### 3.3 SORTING-BASED

We note that efficient sorting algorithms, such as Quicksort and Heapsort, depend on pairwise comparisons. We can use the pairwise preferences from LLMs as the comparator for sorting algorithms. We use Heapsort in this paper due to its guaranteed $O(N \log N)$ computation complexity. We call this approach PRP-Sorting.

PRP-Sorting favors lower computation complexity than PRP-Allpair while also being large insensitive to input orders. However, since it performs local comparisons/swaps on-the-fly and pairwise comparisons are not guaranteed to be transitive, its performance needs to be empirically evaluated.

### 3.4 SLIDING WINDOW

We introduce a sliding window approach that is able to further bring down the computation complexity. One sliding window pass is similar to one pass in the Bubble Sort algorithm: Given an initial ranking, we start from the bottom of the list, compare and swap document pairs with a stride of 1 on-the-fly based on LLM outputs. One pass only requires $O(N)$ time complexity. See Figure 3 for an illustration.

By noticing that ranking usually only cares about Top-$K$ ranking metrics, where $K$ is small, we can perform $K$ passes. For $N = 100$ and $K = 10$, it still only requires 10% LLM API calls of the PRP-Allpair. We call this approach PRP-Sliding-K.

PRP-Sliding-K has favorable time complexity but may have high dependency on input order. In experiments we show surprisingly good results with PRP-Sliding-10, without being very sensitive to input ordering empirically (Appendix A).

### 3.5 REMARKS

In this work, we focus on open-sourced LLMs that are easily accessible to academic researchers, and do not require inquiry of commercial LLM APIs, alleviating some monetary constraints. Also, the LLMs do not need to be finetuned in the prompting-based setting. However, we do acknowledge the cost to prompting LLMs in general.

Here we briefly summarize the properties of pointwise, pairwise, and listwise ranking promptings in Table 1, showing pairwise ranking prompting has several favorable properties.

Table 1: Comparison of pointwise, listwise, and pairwise approaches. $N$ is the number of documents to be ranked for each query. $O(N)$ for listwise approach is based on sliding window since other options are not practical. "Require Calibration" means LLMs need accurate probability estimation to do good ranking, see Section 2.1. $K$ is in terms of top-$K$ ranking metrics that is usually small (<=10) even if thousands of items are being ranked (Zhuang et al., 2023).

| Method | # of LLM API Calls | Generation API | Scoring API | Require Calibration |
|---|---|---|---|---|
| Pointwise | $O(N)$ | No | Yes | Yes |
| Listwise | $O(N)$ | Yes | No | No |
| Pairwise | $O(N^2), O(N \log N), O(KN)$ | Yes | Yes | No |

## 4 EXPERIMENTS ON TREC DL DATASETS

### 4.1 DATASETS AND METRICS

TREC is a widely used benchmark dataset in information retrieval research. We use the test sets of the 2019 and 2020 competitions: TREC-DL2019 and TREC-DL2020, which provide dense human relevance annotations for each of their 43 and 54 queries. Both use the MS MARCO v1 passage corpus, which contains 8.8 million passages. All comparisons are based on the reranking of top 100 passages retrieved by BM25 (Lin et al., 2021) for each query. This is the same setting as existing work (Sun et al., 2023; Ma et al., 2023).

### 4.2 METHODS

We evaluate PRP variants based on open-sourced LLMs, including FLAN-T5-XL, FLAN-T5-XXL (Chung et al., 2022), and FLAN-UL2 (Tay et al., 2022a), which have significantly smaller model sizes (3B, 11B, 20B) than alternatives, and are easily accessible to academic researchers. We report PRP variants including PRP-Allpair, PRP-Sorting, and PRP-Sliding-K.

We consider the following supervised baselines, all trained on the in-domain MS MARCO dataset:

- monoBERT (Nogueira & Cho, 2019): A cross-encoder re-ranker based on BERT-large.
- monoT5 (Nogueira et al., 2020): A sequence-to-sequence re-ranker that uses T5 to calculate the relevance score with pointwise ranking loss.
- RankT5 (Zhuang et al., 2023): A re-ranker that uses T5 and listwise ranking loss.

We also consider the following unsupervised LLM-based baselines:

- Unsupervied Passage Re-ranker (UPR) (Sachan et al., 2022): The *pointwise* approach based on query generation, see Section 2.1.
- Relevance Generation (RG) (Liang et al., 2022): The *pointwise* approach based on relevance generation, see Section 2.1.
- RankGPT (Sun et al., 2023): The *listwise* prompting based approach using various GPT based LLMs. As discussed in Section 2.2, we tried the proposed listwise prompt on FLAN-T5 and FLAN-UL2 models and the outputs are not usable, so we only report results with large blackbox LLMs as in their paper.
- Listwise Reranker with a Large language model (LRL) (Ma et al., 2023): A similar approach to RankGPT with slightly different prompt design.

### 4.3 MAIN RESULTS

Our main results are shown in Table 2. Overall we are able to achieve very encouraging results using PRP. We have the following observations:

- PRP variants based on FLAN-UL2 with 20B parameters can achieve best results on all metrics on TREC-DL2020, and are only second to the blackbox, commercial gpt-4 based solution on NDCG@5 and NDCG@10 on TREC-DL2019, which has an estimated 50X

Table 2: Results on TREC-DL2019 and TREC-DL2020 datasets by reranking top 100 documents retrieved by BM25. Best overall model is in boldface, best and second best unsupervised LLM method are underlined and italicized respectively, for each metric. All unsupervised LLM methods use BM25 to resolve prediction conflicts or failures. *OpenAI has not publicly released the model parameters and the numbers are based on public estimates (VanBuskirk, 2023; Baktash & Dawodi, 2023)

| Method | LLM | Size | TREC-DL2019 | | | TREC-DL2020 | | |
|---|---|---|---|---|---|---|---|---|
| | | | NDCG@1 | NDCG@5 | NDCG@10 | NDCG@1 | NDCG@5 | NDCG@10 |
| BM25 | NA | NA | 54.26 | 52.78 | 50.58 | 57.72 | 50.67 | 47.96 |
| *Supervised Methods* | | | | | | | | |
| monoBERT | BERT | 340M | 79.07 | 73.25 | 70.50 | 78.70 | 70.74 | 67.28 |
| monoT5 | T5 | 220M | 79.84 | 73.77 | 71.48 | 77.47 | 69.40 | 66.99 |
| monoT5 | T5 | 3B | 79.07 | 73.74 | 71.83 | 80.25 | 72.32 | 68.89 |
| RankT5 | T5 | 3B | 79.07 | 75.66 | 72.95 | 80.86 | 73.05 | 69.63 |
| *Unsupervised LLM Methods* | | | | | | | | |
| LRL | text-davinci-003 | 175B | - | - | 65.80 | - | - | 62.24 |
| RankGPT | gpt-3 | 175B | 50.78 | 50.77 | 49.76 | 50.00 | 48.36 | 48.73 |
| RankGPT | text-davinci-003 | 175B | 69.77 | 64.73 | 61.50 | 69.75 | 58.76 | 57.05 |
| RankGPT | gpt-3.5-turbo | 154B* | *82.17* | 71.15 | 65.80 | 79.32 | 66.76 | 62.91 |
| RankGPT | gpt-4 | 1T* | **82.56** | **79.16** | **75.59** | 78.40 | 74.11 | *70.56* |
| UPR | FLAN-T5-XXL | 11B | 62.79 | 62.07 | 62.00 | 64.20 | 62.05 | 60.34 |
| RG | FLAN-T5-XXL | 11B | 67.05 | 65.41 | 64.48 | 65.74 | 66.40 | 62.58 |
| UPR | FLAN-UL2 | 20B | 53.10 | 57.68 | 58.95 | 64.81 | 61.50 | 60.02 |
| RG | FLAN-UL2 | 20B | 70.93 | 66.81 | 64.61 | 75.62 | 66.85 | 65.39 |
| PRP-Allpair | FLAN-T5-XL | 3B | 74.03 | 71.73 | 69.75 | 79.01 | 72.22 | 68.12 |
| PRP-Sorting | FLAN-T5-XL | 3B | 77.52 | 71.88 | 69.28 | 74.38 | 69.44 | 65.87 |
| PRP-Sliding-10 | FLAN-T5-XL | 3B | 75.58 | 71.23 | 68.66 | 75.62 | 69.00 | 66.59 |
| PRP-Allpair | FLAN-T5-XXL | 11B | 72.09 | 71.28 | 69.87 | 82.41 | 74.16 | 69.85 |
| PRP-Sorting | FLAN-T5-XXL | 11B | 74.42 | 69.62 | 67.81 | 72.53 | 71.28 | 67.77 |
| PRP-Sliding-10 | FLAN-T5-XXL | 11B | 64.73 | 69.49 | 67.00 | 75.00 | 70.76 | 67.35 |
| PRP-Allpair | FLAN-UL2 | 20B | 73.64 | 74.77 | 72.42 | *85.19* | *74.73* | **70.68** |
| PRP-Sorting | FLAN-UL2 | 20B | 74.42 | 73.60 | 71.88 | 84.57 | 72.52 | 69.43 |
| PRP-Sliding-10 | FLAN-UL2 | 20B | 78.29 | *75.49* | *72.65* | **85.80** | **75.35** | 70.46 |

larger model size. Our best methods outperform RankGPT based on text-davinci-003 with 175B parameters by over 10% on all ranking metrics, and are competitive to supervised methods on all ranking metrics.

- Results on FLAN-T5-XL and FLAN-T5-XXL are also competitive, showing that PRP generalizes to smaller LLMs due to the significant simplicity of the pairwise ranking comparisons. They generally work even better than the gpt-3.5.turbo based solution (10X - 50X in size) on the more stable NDCG@5 and NDCG@10 metrics, and outperforms text-davinci-003 based solution on all ranking metrics.

- It is encouraging to see good results from efficient PRP variants. For example, the sliding window variants generally get very robust ranking performance and we get some of the best metrics from this variant. This observation alleviates some efficiency concerns of pairwise ranking approaches.

## 5 EXPERIMENTS ON BEIR DATASETS

### 5.1 DATASETS AND METRICS

BEIR (Thakur et al., 2021) consists of diverse retrieval tasks and domains. Following (Sun et al., 2023) we choose the test sets of Covid, Touche, DBPedia, SciFact, Signal, News, and Robust04. Following the convention of related research, we report NDCG@10 for each dataset and the average NDCG@10.

### 5.2 METHODS

We use the same prompt template from TREC datasets for all BEIR datasets, which is consistent for all compared unsupervised LLM-based baselines. This is in contrast to methods such as (Dai et al.,

Table 3: Results (NDCG@10) on BEIR datasets. All models re-rank the same BM25 top-100 passages. Best overall model is in boldface, best and second best unsupervised LLM method are underlined and italicized respectively, for each metric. All unsupervised LLM methods use BM25 to resolve prediction conflicts or failures.

| Method | LLM | Size | Covid | Touche | DBPedia | SciFact | Signal | News | Robust04 | Avg |
|---|---|---|---|---|---|---|---|---|---|---|
| BM25 | NA | NA | 59.47 | **44.22** | 31.80 | 67.89 | 33.05 | 39.52 | 40.70 | 45.23 |
| **Supervised Methods** | | | | | | | | | | |
| monoBERT | BERT | 340M | 70.01 | 31.75 | 41.87 | 71.36 | 31.44 | 44.62 | 49.35 | 48.63 |
| monoT5 | T5 | 220M | 78.34 | 30.82 | 42.42 | 73.40 | 31.67 | 46.83 | 51.72 | 50.74 |
| monoT5 | T5 | 3B | 80.71 | 32.41 | 44.45 | **76.57** | 32.55 | 48.49 | 56.71 | 53.13 |
| RankT5 | T5 | 3B | 82.00 | 37.62 | 44.19 | 76.86 | 31.80 | 48.15 | 52.76 | 53.34 |
| TART-Rerank | T5 | 3B | 75.10 | 27.46 | 42.53 | 74.84 | 25.84 | 40.01 | 50.75 | 48.08 |
| **Unsupervised LLM Methods** | | | | | | | | | | |
| UPR | FLAN-T5-XXL | 11B | 72.64 | 21.56 | 35.14 | 73.54 | 30.81 | 42.99 | 47.85 | 46.36 |
| RG | FLAN-T5-XXL | 11B | 70.31 | 22.10 | 31.32 | 63.43 | 26.89 | 37.34 | 51.56 | 43.28 |
| UPR | FLAN-UL2 | 20B | 70.69 | 23.68 | 34.64 | 71.09 | 30.33 | 41.78 | 47.52 | 45.68 |
| RG | FLAN-UL2 | 20B | 70.22 | 24.67 | 30.56 | 64.74 | 29.68 | 43.78 | 53.00 | 45.24 |
| RankGPT | gpt-3.5-turbo | 154B | 76.67 | 36.18 | 44.47 | 70.43 | 32.12 | *48.85* | 50.62 | 51.33 |
| PRP-Allpair | FLAN-T5-XL | 3B | 81.86 | 26.93 | 44.63 | 73.25 | 32.08 | 46.52 | 54.02 | 51.33 |
| PRP-Sorting | FLAN-T5-XL | 3B | 80.41 | 28.23 | 42.84 | 67.94 | 30.95 | 42.95 | 50.07 | 49.06 |
| PRP-Sliding-10 | FLAN-T5-XL | 3B | 77.58 | *40.48* | 44.77 | 73.43 | **35.62** | 46.45 | 50.74 | 52.72 |
| PRP-Allpair | FLAN-T5-XXL | 11B | 79.62 | 29.81 | 41.41 | *74.23* | 32.22 | 47.68 | **56.76** | 51.67 |
| PRP-Sorting | FLAN-T5-XXL | 11B | 78.75 | 29.61 | 39.23 | 70.10 | 31.28 | 44.68 | 53.01 | 49.52 |
| PRP-Sliding-10 | FLAN-T5-XXL | 11B | 74.39 | 41.60 | 42.19 | 72.46 | 35.12 | 47.26 | 52.38 | 52.20 |
| PRP-Allpair | FLAN-UL2 | 20B | **82.30** | 29.71 | *45.94* | *75.70* | 32.26 | 48.04 | *55.49* | *52.78* |
| PRP-Sorting | FLAN-UL2 | 20B | *82.29* | 25.80 | 44.53 | 67.07 | 32.04 | 45.37 | 51.45 | 49.79 |
| PRP-Sliding-10 | FLAN-UL2 | 20B | 79.45 | 37.89 | **46.47** | 73.33 | *35.20* | **49.11** | 53.43 | **53.55** |

2022) that require prior knowledge to design different prompts for different datasets, which may be difficult in practice and will lead to unfair comparisons.

For supervised methods, in addition to the baselines in Section 4.2, we add TART (Asai et al., 2023), a supervised instruction-tuned passage re-ranker trained on 37 datasets, including over 5 million instances. The model is initialized from FLAN-T5-XL.

For unsupervised LLM methods, we also report RG and UPR as in Section 4.2. We include RankGPT with gpt-3.5-turbo. We do not include the GPT-4 numbers reported in (Sun et al., 2023), which used GPT-4 to *rerank* top results from gpt-3.5-turbo due to the significant cost. It essentially performed an ensemble of two re-ranking models, which is unfair and impractical. We also do not include LRL since it was not evaluated on the BEIR collection. See more discussions of baselines in Appendix E.

## 5.3 MAIN RESULTS

The main results are shown in Table 3. Overall we are able to achieve encouraging results using PRP, validating its robustness across different domains. We have the following observations:

- PRP variants based on FLAN-UL2 with 20B parameters can achieve best overall results on the collection.

- PRP variants generate the best ranking metrics on all datasets among unsupervised LLM methods. PRP outperforms the blackbox commercial RankGPT solution by 4.2%, and pointwise LLM-based solutions by over 10% in general. Noticably, PRP-Sliding-10 with FLAN-UL2 outperforms RankGPT on *all* 7 datasets, showing its strong generalization.

- PRP performs favorably with supervised methods. PRP-Sliding-10 with FLAN-UL2 can slightly outperform the state-of-the-art RankT5 ranker on average, and outperform RankT5 on 5 out of 7 datasets.

- Results on FLAN-T5-XL and FLAN-T5-XXL are again competitive, some variants can outperform RankGPT.

## 5.4 ABLATION STUDIES

We perform several ablative studies to gain a deeper understanding of the PRP framework. We show the robustness of PRP to input ordering in Appendix A, the applicability of PRP for both generation and scoring API in Appendix B, and provide more study on the sliding window approach in Appendix C.

## 6 DISCUSSION

The design of PRP in this paper biases towards simplicity and generality, and the performance may further improve via more sophisticated prompt design, and leveraging extra information such as the score values from scoring API (which will then not be applicable to generation API).

We note there is no label leakage issues as we leverage open-sourced LLMs with clear documentations, while it is not clear for blackbox commercial LLMs. Please see more discussions on limitations and future work of PRP in Appendix D.

## 7 RELATED WORK

We did a detailed review and analysis of the most relevant existing efforts for ranking with LLMs, including pointwise and listwise approaches in Section 2. These works and ours focus on the challenging unsupervised text ranking setting with LLMs without providing any examplers, conducting any fine-tuning, or training of an additional model. Prior to the recent efforts related to ranking with LLMs, most work focus on the supervised learning to rank problem (Liu, 2009; Qin et al., 2021) by fine-tuning Pre-trained Language Models (PLMs) such as T5 (Nogueira et al., 2020; Zhuang et al., 2023) or BERT (Nogueira & Cho, 2019; Zhuang et al., 2021), which serve as very strong baselines.

There has been a strong recent interest in exploring information retrieval in general with LLMs based approaches (Zhu et al., 2023), due to the importance of the applications and the power of LLMs to understand textual queries and documents (Dai et al., 2022; Tay et al., 2022b; Wang et al., 2023; Jagerman et al., 2023; Bonifacio et al., 2022). Several works leverage the generation power of LLMs to generate training data to train an additional downstream retrieval or ranking model, typically in the few-shot setting (Dai et al., 2022), which is a very different setting from ours. Recent methods in this family of methods such as Inpars (Bonifacio et al., 2022) still significantly underperforms fine-tuned baselines. ExaRanker (Ferraretto et al., 2023) uses LLMs to generate explanations for ranking decisions, and uses such explanations in ranking model fine-tuning, showing limited ranking performance benefits (the major benefit was on data efficiency). HyDE (Gao et al., 2022) uses LLMs to augment queries by generating hypothetical documents for unsupervised retrieval. These works do not directly explore the retrieval or ranking capability of LLMs, but mainly use LLMs as auxiliary tools to complement traditional paradigms, possibly limiting the benefits that LLMs can provide. New paradigms such as Differentiable Search Index (DSI) (Tay et al., 2022b; Wang et al., 2022) directly use Transformer memory to index documents for retrieval. Though novel, they mainly focus on retrieval, and the performance gap from supervised baselines is still large.

Using pairwise comparisons with LLMs is a general paradigm, such as reward modeling using pairwise preferences (Christiano et al., 2017). LLMs are used as evaluators to compare generative outputs (such as text summary) (Liu et al., 2023b). 1SL (MacAvaney & Soldaini, 2023) estimates relevance with reference to an anchor positive query-document pair *per query*, even for the test set, so the setting may not be practical and is very different from our standard text ranking setting. A concurrent work (Dai et al., 2023) studied pairwise prompting in recommender systems, showing impressive performance especially in cold-start settings. However, it is a substantially different application and their method still fall behind state-of-the-art models with sufficient data. The novelty of our work lies in leveraging the general and simple pairwise prompting paradigm to the important text ranking task, granting LLMs capabilities that no prior work can, by performing competitively with state-of-the-art fine-tuned models and methods that only work with giant blackbox LLMs.

## 8 CONCLUSION

In this paper, we propose to use pairwise prompting with LLMs for text ranking tasks. To the best of our knowledge, these are the first published results demonstrating very competitive ranking performance using moderate-sized, open-sourced LLMs. The key insights are the observation of the difficulties of LLMs handling ranking tasks in the existing pointwise and listwise formulations. Our proposed Pairwise Ranking Prompting (PRP) is effective in reducing the burden of LLMs and shows robust performance on 9 datasets. We also discuss efficiency concerns and ways to mitigate them, and several benefits of PRP, such as insensitivity to input ordering and support for both generation and scoring LLM APIs.

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

APPENDIX

## A ROBUSTNESS TO INPUT ORDERING

One issue of listwise ranking prompting approaches is their sensitivity to input ordering. This is because the ranking will fall back to the initial order when LLM prediction fails, which is very common for the difficult listwise formulation. In Table 4 we show results of different methods by inverting the initial order from BM25.

Table 4: Input order sensitivity results on the TREC-DL2019 dataset.

| Method | LLM | Init Order | NDCG@1 | NDCG@5 | NDCG@10 |
|---|---|---|---|---|---|
| RankGPT | gpt-3.5-turbo | BM25 | 82.17 | 71.15 | 65.80 |
| RankGPT | gpt-3.5-turbo | Inverse BM25 | 36.43 | 31.79 | 32.77 |
| PRP-Allpair | FLAN-UL2-20B | BM25 | 73.64 | 74.77 | 72.42 |
| PRP-Allpair | FLAN-UL2-20B | Inverse BM25 | 74.42 | 74.48 | 72.40 |
| PRP-Sliding-1 | FLAN-UL2-20B | BM25 | 78.29 | 62.15 | 57.58 |
| PRP-Sliding-1 | FLAN-UL2-20B | Inverse BM25 | 71.32 | 32.72 | 26.04 |
| PRP-Sliding-10 | FLAN-UL2-20B | BM25 | 78.29 | 75.49 | 72.65 |
| PRP-Sliding-10 | FLAN-UL2-20B | Inverse BM25 | 71.32 | 67.91 | 64.84 |

As expected, PRP-Allpair is quite robust to initial ordering, and PRP-Sliding-1 will suffer for metrics other than NDCG@1. PRP-Sliding-10 is quite robust since it focuses on Top-K ranking metrics.

## B COMPARISON OF SCORING MODE AND GENERATION MODE

Our results above are all based on the scoring mode, since PRP only need to get scores for two candidate outputs ("Passage A" and "Passage B") and it is easy to get probabilities from open-sourced LLMs. Here we compare against PRP performance using scoring vs generation mode in Table 5, which will shed light on how PRP works on generation-only LLM APIs.

Table 5: Results on TREC-DL2019 and TREC-DL2020 datasets using scoring vs generation mode for PRP.

| Method | LLM | Mode | TREC-DL2019 | | | TREC-DL2020 | | |
|---|---|---|---|---|---|---|---|---|
| | | | NDCG@1 | NDCG@5 | NDCG@10 | NDCG@1 | NDCG@5 | NDCG@10 |
| PRP-Allpair | FLAN-T5-XL | Scoring | 74.03 | 71.73 | 69.75 | 79.01 | 72.22 | 68.12 |
| PRP-Allpair | FLAN-T5-XL | Generation | 74.03 | 71.68 | 69.59 | 79.01 | 71.54 | 67.75 |
| PRP-Allpair | FLAN-T5-XXL | Scoring | 72.09 | 71.28 | 69.87 | 82.41 | 74.16 | 69.85 |
| PRP-Allpair | FLAN-T5-XXL | Generation | 72.09 | 71.61 | 69.94 | 80.56 | 73.69 | 69.53 |
| PRP-Allpair | FLAN-UL2 | Scoring | 73.64 | 74.77 | 72.42 | 85.19 | 74.73 | 70.68 |
| PRP-Allpair | FLAN-UL2 | Generation | 73.64 | 74.84 | 72.37 | 85.19 | 74.74 | 70.69 |

We can see that PRP is extremely robust to scoring vs generation API, even for smaller LLMs, showing its applicability to different LLMs systems. The results are intuitive - LLMs make few generation mistakes due to the simplicity of PRP. We found that there are only about 0.02% predictions that do not follow the desired format, which is neglectable and in stark contrast to the the listwise approaches.

## C MORE RESULTS ON PRP-SLIDING-K

We show more results on PRP-Sliding-K variants to better understand the behaviors, including multiple backward passes and a forward pass variant[1]. The results are shown in Table 6 and Table 7 on TREC-DL2019 and TREC-DL2020 with consistent behaviors.

The results are easy to interpret:

- The behavior is similar to BubbleSort: Strong NDCG@1 can already be achieved with one backward pass. As we conduct more passes, other Top-K ranking metrics get better.

---

[1]Backward pass indicates starting from the bottom result with the lowest BM25 score, and vice versa.

Table 6: Sliding window results on the TREC-DL2019 dataset.

| Method | LLM | Strategy | NDCG@1 | NDCG@5 | NDCG@10 |
|--------|-----|----------|--------|--------|---------|
| PRP-Sliding | FLAN-UL2-20B | 1 Forward | 63.95 | 57.31 | 54.10 |
| PRP-Sliding | FLAN-UL2-20B | 1 Backward | 78.29 | 62.15 | 57.58 |
| PRP-Sliding | FLAN-UL2-20B | 2 Backward | 78.29 | 67.01 | 61.52 |
| PRP-Sliding | FLAN-UL2-20B | 3 Backward | 78.29 | 70.72 | 64.60 |
| PRP-Sliding | FLAN-UL2-20B | 10 Backward | 78.29 | 75.49 | 72.65 |

Table 7: Sliding window results on the TREC-DL2020 dataset.

| Method | LLM | Strategy | NDCG@1 | NDCG@5 | NDCG@10 |
|--------|-----|----------|--------|--------|---------|
| PRP-Sliding | FLAN-UL2-20B | 1 Forward | 65.74 | 54.72 | 51.21 |
| PRP-Sliding | FLAN-UL2-20B | 1 Backward | 85.80 | 61.60 | 57.06 |
| PRP-Sliding | FLAN-UL2-20B | 2 Backward | 85.80 | 66.51 | 61.11 |
| PRP-Sliding | FLAN-UL2-20B | 3 Backward | 85.80 | 71.06 | 63.45 |
| PRP-Sliding | FLAN-UL2-20B | 10 Backward | 85.80 | 75.35 | 70.46 |

- Forward pass does not work well, which is intuitive, since it mainly performs demotion and is much less efficient in bringing good results to the top.

## D    MORE DISCUSSION ON LIMITATIONS AND FUTURE WORK

**Cost and Efficiency.**    We discussed different efficient variants of PRP. Also, our results are based on LLMs that are easily approachable for academic researchers (Taori et al., 2023), alleviating the need to call commercial APIs. However, further reducing the number of calls to LLMs is still an interesting research direction, such as leveraging active learning techniques.

**Domain adaptation.**    The datasets used in this paper are for the standard and important relevance-based text ranking. How LLMs can be adapted to non-standard ranking datasets, such as counter arguments in the ArguAna dataset (Wachsmuth et al., 2018), need more investigation. Our work can facilitate such explorations by providing approachable baselines.

**Other Models.**    We do not use GPT models (though we compare with them using results from other papers) in this work due to various constraints. Testing the performance of our methods on such models is meaningful benchmarking effort.

**Ranking-aware LLMs.**    We, as other existing work, focus on unsupervised ranking with off-the-shelf LLMs, and show that pairwise ranking is the ideal prompting unit. How to make LLMs more ranking-aware, in a data efficient manner, while maintaining their generality for other tasks, is a challenging research direction.

**Data leakage.**    We mainly use open-sourced FLAN models (Wei et al., 2021) with clear documentations, which neither observed ranking supervision from any of the datasets we evaluated upon, nor was instruction fine-tuned on any ranking tasks. Also, the labels in the datasets are *dense* human annotations for each query against many documents, which are not used in the open-sourced LLMs and are very different from the potential usage of document corpus during pre-training. These are in contrast to methods based blackbox LLMs such as ChatGPT or GPT-4 (Sun et al., 2023) where the tuning details are unclear. We do note that FLAN models have a question answering task based on MSMARCO, which is not ranking specific, and is different from TREC-DL datasets in terms of queries and annotations, and is different from BEIR collection in all aspects. On the other hand, whether blackbox LLMs directly use TREC-DL datasets or BEIR datasets is unclear.

## E    MORE DISCUSSION ON BASELINE AND DATASET SELECTION

For the BEIR evaluation, we choose not to include the Promptagator++ ranker (Dai et al., 2022) since 1) It uses different prompts and fine-tuned models for each task, different from all other LLM

methods. 2) The method was evaluated on a different set of BEIR tasks. Even for the shared tasks, it reranks top 200 results from a stronger retriever than BM25 so the numbers are not comparable. Nevertheless, zero-shot Promptagator++ performed significantly *worse* than the monoT5 baseline in the paper (to be fair, the paper's focus was mainly on few-shot scenarios), while PRP compares favorably with monoT5.

The only dataset we did not include, but (Sun et al., 2023) included, from the BEIR collection, is the NFCorpus dataset. This is because the metrics using BM25 reported in (Sun et al., 2023) on NFCorpus does not match ours and the public consensus numbers (while the numbers match for all selected datasets), so we exclude NFCorpus to avoid unfair comparisons possibly due to errors during their evaluation.

## F  REPRODUCIBILITY

### F.1  PAIRWISE RANKING PROMPTING TEMPLATE

We note that we used the **same prompt template for all 9 datasets** evaluated in the paper, showing the generality and power of pairwise ranking prompting in text ranking. Below is the prompt template:

> Given a query {query}, which of the following two passages is more relevant to the query?
>
> Passage A: {document$_1$}
>
> Passage B: {document$_2$}
>
> Output Passage A or Passage B:

### F.2  CODE AND DATA RELEASE

As we focus on open-sourced LLMs, and only use standard aggregation methods (win counting, sorting, and sliding window), our experimental results are easy to reproduce. We plan to release the code (including the prompt and the rank aggregation functions). Further we plan to release pairwise inference results on all 9 datasets and the 3 open-source LLMs to facilitate future research. In specific, we will release the data in the following json format, which includes query/document information for each pair (including ids, text, label, retrieval rank and scores), together with the actual prompt, the generated text, and its score. Below is an example on the Trec-DL2020 dataset with Flan-UL2:

"document_pair": [{"document_id": "8512412", "retriever_rank": "50", "retriever_score": "8.984600", "document": "When in Doubt, Take a Cab. Taxis might be expensive in Puerto Rico, but they are safe and available. At night, it's definitely the best way to get around. Look for the white taxis with the distinctive garita, or sentry box, icon painted on them.They are usually found at designated taxi stands.hen in Doubt, Take a Cab. Taxis might be expensive in Puerto Rico, but they are safe and available. At night, it's definitely the best way to get around. Look for the white taxis with the distinctive garita, or sentry box, icon painted on them.", "relevance": -1}, {"document_id": "6623205", "retriever_rank": "66", "retriever_score": "8.812100", "document": "Thankfully, there are a couple of ways to prevent your whites from turning yellow: 1 Never bleach white clothing that is polyester or a polyester/cotton blend. 2 The chemical reaction between the bleach and the polyester almost always yields a yellowed result. 3 Consider a water softener if you have well-water.hankfully, there are a couple of ways to prevent your whites from turning yellow: 1 Never bleach white clothing that is polyester or a polyester/cotton blend. 2 Consider a water softener if you have well-water. 3 Minimize your use of bleach altogether.", "relevance": 1.0}],

"query_id": "1108651",

"query": "what the best way to get clothes white",

"prompt": "Given a query "what the best way to get clothes white", which of the following two passages is more relevant to the query?

Passage A: When in Doubt, Take a Cab. Taxis might be expensive in Puerto Rico, but they are safe and available. At night, it's definitely the best way to get around. Look for the white taxis with the distinctive garita, or sentry box, icon painted on them.They are usually found at designated taxi stands.hen in Doubt, Take a Cab. Taxis might be expensive in Puerto Rico, but they are safe and available. At night, it's definitely the best way to get around. Look for the white taxis with the distinctive garita, or sentry box, icon painted on them.

Passage B: Thankfully, there are a couple of ways to prevent your whites from turning yellow: 1 Never bleach white clothing that is polyester or a polyester/cotton blend. 2 The chemical reaction between the bleach and the polyester almost always yields a yellowed result. 3 Consider a water softener if you have well-water.hankfully, there are a couple of ways to prevent your whites from turning yellow: 1 Never bleach white clothing that is polyester or a polyester/cotton blend. 2 Consider a water softener if you have well-water. 3 Minimize your use of bleach altogether.

Output Passage A or Passage B:",

"generated_text": "Passage B",

"prediction_score": -0.0025123630184680223

