# OpenReview forum: "Large Language Models are Effective Text Rankers with Pairwise Ranking Prompting"
_ICLR.cc/2024/Conference — ICLR 2024 Conference Withdrawn Submission_

### Official Review · Reviewer_2oVm · 2023-11-01

**Soundness:** 3 good
**Presentation:** 3 good
**Contribution:** 3 good
**Rating:** 5
**Confidence:** 3

**Summary:**

This paper tackles an important problem of applying unsupervised LLMs for document ranking. Authors quickly pointed out severe limitations of existing LLM-powered ranking methods. One class of such methods (pointwise) requires a calibrated score (or probability) which would be difficult to obtain between different prompts. The second class of ranking methods (listwise) suffers from issues ranging from missing expected ranked items to repetitions and various forms of inconsistency.
To address those issues authors presented a variant of a well-known (in learning-to-rank literature) pairwise ranking method adapted for  unsupervised LLMs for ranking documents.
Besides simplicity the main benefits of this method is two-fold. First, carefully calibrated scores are not required since only two documents are being scored at a time and relative scores are sufficient. Second, only a fraction of expensive LLM API calls are needed. Additionally, the authors used relatively-small (still in billions) open source LLM with a smaller number of parameters and produced a very competitive results agains a very large black box LLMs and also against supervised methods when tested on widely used information retrieval datasets.

**Strengths:**

This paper is original in a sense that the authors found real problems with the existing state-of-the art in LLM-assisted ranking methods and proposed a method that appears to solve all the main issues they identified with the existing methods.
The focus of the work was not only on producing a better ranking method but also in decreasing the cost of final rankings. LLM API calls are expensive, hosting a very large LLM can be prohibitevly expensive.
The authors addressed these problem by choosing a much smaller (in terms of parameters) open-source LLM compared to very large black-box LLMs (chatGPT variants). They also proposed several methods with decreasing number of LLM calls (allpairst, sorting-based and sliding-window)
The authors were also careful to point out that only 0.02% predictions did not follow desired format when generating output. I am assuming that the number of generation errors among listwise methods is considerably worse.
And the results of running their method on widely used datasets were very competitive with existing ranking methods.

**Weaknesses:**

I would like to raise a couple of issues.
The first issue is that the authors have not shared their source code. They did mentioned the type of open-source LLM they were using but it would not be easy for somebody short-on-time to quickly reporoduce their results. I think it would be useful if the authors shared the actual prompts they used and the code they used for computing their results.

While reading this paper I realized that this paper would definitely benefit from the discussion on the following statement from the paper:
"...However, since it performs local comparisons/swaps on-the-fly and pairwise comparisons are not guaranteed to be transitive, its performance needs to be empirically evaluated..."

When comparisons are not guaranteed to be transitive, we are not working within the usual framework of a total order. I believe traditional sorting algorithms assume some transitivity.
So not only the average runtime complexity may be affected but validity of the output as well.
Specifically for HeapSort we may end up with violating of the heap property if our ordering is not transitive since even if both children are less than their parent, we can't guarantee that the the grandchild is less than the grandparent and vice versa, breaking the heap property and the ability to maintain it.

**Questions:**

How the absence of the assumption of the total order would affect the the total number of API calls (reported in Table 1)

---

> ### Author Response · Authors · 2023-11-15
> **Thank you for your review**
>
> We really appreciate your in-depth review.
>
> We want to clarify the reproducibility of this work. We added a “Reproducibility” section in the appendix of the updated version of the paper. In particular, we plan to open-source the code, which is simple, as we use the *exact same prompt in Figure 2* for all 9 datasets, showing the generality of the paradigm. As stated, we also plan to release the pairwise predictions for all pairs on all datasets we tested on, so researchers do not need to run the LLM inferences. Please see more discussions in the updated version of the paper.
>
>
> In terms of "...However, since it performs local comparisons/swaps on-the-fly and pairwise comparisons are not guaranteed to be transitive, its performance needs to be empirically evaluated...".  We appreciate your careful comment on the sorting-based algorithm.
>
> We note that it is unrealistic to expect a (LLM-based) pairwise ranker to always make correct predictions, because without knowing the ground-truth relevance of each document it is bound to be noisy. What most pairwise rankers can do is to be as "probably correct" as possible.
>
> When the ranker is not 100% correct, yes, the ranking is not guaranteed to be 100% correct as well. However, an  "approximately correct" ranking is still valuable, and a "probably correct" pairwise ranker with sliding window / heap sort is still likely to deliver an "approximately correct" ranking, as demonstrated empirically in Table 2 and Table 3 on a wide range of benchmark datasets. We agree that a more theoretically rigorous proof of transitivity would be a nice direction for future work, but is out of scope of the current paper.

---

> > ### Author Response · Authors · 2023-11-20
> >
> > We wonder if our responses address your. concerns, thank you.

---

> > > ### Comment · Reviewer_2oVm · 2023-11-23
> > >
> > > Thank you for your response to my review.
> > > I really appreciate that you added a section on reproducibility in the appendix and I understand your point about pairwise rankers being "probably correct". Still, I would prefer to see the actual code for the work submitted and regarding worst case time complexity in terms of number of API calls in my opinion the paper would benefit from a more thorough discussion with relevant citations especially since you don't plan on providing rigorous treatment of the subject.
> > > I will keep my original rating.

---

### Official Review · Reviewer_dX3y · 2023-11-06

**Soundness:** 3 good
**Presentation:** 3 good
**Contribution:** 2 fair
**Rating:** 5
**Confidence:** 3

**Summary:**

The paper proposes a pair-wise prompting strategy for unsupervised information retrieval using large language models. The author(s) first analyze the drawbacks of point-wise or list-wise ranking w/ LLMs, then propose two paradigms to call LLMs on pair-wise ranking, namely scoring mode and generation mode. To alleviate the efficiency issue, the author(s) use efficient ranking algorithms or sliding windows to accelerate the whole ranking process. Extensive experiments are conducted on two public benchmarks. Surprisingly, open-source LLMs w/ the proposed pair-wise prompting method can achieve comparable or better results compared to GPT-4.

**Strengths:**

1. The paper studies an important task, i.e., information retrieval.
2. Timely study on information retrieval using large language models.
3. Extensive experiments are conducted on two public benchmarks.
4. Surprisingly results of ranking w/ open-source LLMs using the proposed pair-wise prompting strategies.

**Weaknesses:**

1. Limited novelty. The main novelty lies on twofolds. (1) pair-wise prompting (2) accelerating w/ sorting or sliding window algorithms.
    1. For pair-wise prompting, it is straightforward to consider such kind of ranking paradigm (given that point-wise, pair-wise, and list-wise are three typical paradigms usually discussed in the field of information retrieval). The pair-wise ranking prompting strategy has also been explored in existing works [1].
    2. The sliding window strategy has also been explored in works about searching using LLMs [2].
    3. As a result, although the proposed method shows promising and strong results, especially using open-source LLMs, the new insights provided by this paper are limited.

2. Code is not available for reproduction, which is crucial for this work because it's based on open-source LLMs.

3. The time complexity of the sliding window strategy could be misleading. In information retrieval tasks, it's common to have a relatively large K like 100. As a result, O(KN) is a more precise time complexity for this approach.

4. It would be better if the author(s) could quantitatively present the efficiency of each approach, e.g., comparing the number of LLM calls.

5. Typo (Minor). The bottom of page 6, "second to the blackbox, commercial gpt-4 based solution on NDCG@5 and NDCG@10". However, according to Table 2, should it be NDCG@{1, 5, 10}?


[1] Dai et al. Uncovering ChatGPT’s Capabilities in Recommender Systems.

[2] Sun et al. Is ChatGPT Good at Search? Investigating Large Language Models as Re-Ranking Agents.

**Questions:**

Please refer to "Weaknesses".

---

> ### Author Response · Authors · 2023-11-15
> **Clarification on factual misunderstandings**
>
> We appreciate the reviewer’s effort to review our manuscript. We want to point out *factual* misunderstanding in the review and hope the novelty, scope, and setting of the work is clear:
>
> Novelty: We respectfully argue there’s significant factual misunderstanding of the novelty of this work. In particular, the reviewer comments that 1) pairwise paradigm is easy to think of. 2) It is studied in “Uncovering ChatGPT’s Capabilities in Recommender Systems”. 3) Sliding window is explored in “Is ChatGPT Good at Search? Investigating Large Language Models as Re-Ranking Agents”.
> - For 1), our work is the first to show compelling ranking performance on the important text relevance ranking task with open-sourced, moderated sized LLMs, which is *only possible with our proposed methodology* and has not been done in the literature. We argue that this is novelty, which *should not be judged in terms of simplicity*.
> - For 2) We argue that the paper you pointed to (“Uncovering ChatGPT’s Capabilities in Recommender Systems”, https://arxiv.org/abs/2305.02182) is concurrent work, which was not presented until RecSys (Sep 2023). We cited it in the updated version and have some discussions in the Related Work section. We argue that this work is *substantially* different from the text relevance ranking task, a critical problem in web search. We discussed in Related Work that “pairwise” is a general paradigm used in different ways for LLMs and argue that novelty should not be judged by general paradigms such as pointwise / pairwise. Also, it is non-trivial to apply pairwise comparisons to practical ranking tasks, where we study various strategies with different trade-offs.
> - For 3) Sliding windows is also a *general* tool to show the proposed method *can* be made more practical, and we *did not claim algorithmic novelty* of it. Pairwise prompting avoids production of unexpected outputs such as repetition and missing from listwise prompting and naturally mitigates prompt length limit and position bias in the prompt. Thus, though RankGPT and ours both leverage sliding windows, their properties are significantly different due to the computation unit - note that the RankGPT listwise prompting is completely unusable with moderate LLMs as discussed in Sec 2.2, even if sliding window is used.
>
> Scope: Another potential misunderstanding we also want to point out that BEIR is a dataset *collection*, and we evaluated 7 datasets from it with very different natures / domains. Also, researchers in this field normally treat TREC DL19/20 as two separate datasets. So we use *nine benchmark datasets, instead of two as indicated in the review*.
>
> Setting: We are not sure if we understand the comment “it's common to have a relatively large K like 100”. For experiments we are re-ranking 100 documents as done in the literature. Sliding-window-K is to optimize Top-K metrics that the search ranking field is concerned with, where K is usually small (<=10), even if more documents are ranked. Also note we name our methods reported in experiments explicitly - i.e., sliding-window-*10*. Nevertheless, we have updated the paper by explicitly acknowledging the complexity is O(KN) in Table 1 and thanks for the suggestion.

---

> ### Author Response · Authors · 2023-11-15
> **We acknowledge other comments that help to improve the paper**
>
> We acknowledge other comments that help to improve the paper.
>
> We added a “Reproducibility” section in the appendix of the updated version of the paper. In particular, we plan to open-source the code that uses the exact same prompt in Figure 2 for all 9 datasets, and the ranking aggregations are based on standard counting, sorting, and sliding-window methods. We also plan to release the pairwise predictions for all pairs on all datasets we tested on, so researchers do not need to run the LLM inferences. Please see more discussions in the updated version of the paper.
>
> In terms of API calls, thanks for the suggestions and we plan to discuss these in the next version of the paper. For a query with 100 documents, AllPair needs ~20k API calls, Sorting needs ~2k API calls, Sliding-Window-10 needs ~2k API calls. Note that when there are more documents, the benefits of Sliding-Window-10 will be even clearer. Also note that our pairwise prompt is significantly shorter than the listwise prompts, so each API call is cheaper. Please see more discussion in section 3.5, where we acknowledge the cost issue in general, but discuss benefits that are only possible with our work. Further reducing the cost is a valid future direction.
>
> For the typo, thanks for pointing this out. We use “second to” to indicate the method is the 2nd best approach. Our methods are inferior to both GPT-4 and ChatGPT for that metric on the particular dataset (i.e., ranks 3rd) so we believe the description is correct, but we will find better ways to express this.

---

> > ### Author Response · Authors · 2023-11-20
> >
> > We wonder if our responses address your. concerns, thank you.

---

> ### Comment · Reviewer_dX3y · 2023-11-22
>
> Thank you for your detailed response and clarification regarding my review.
>
> 1. Regarding the discussion on novelty, personally, I do not recognize the discussion on novelty as something of a factual misunderstanding. As I mentioned under "Strengths" in my initial review, I highly appreciate the strong performance demonstrated by the proposed method, particularly with the use of open-source LLMs. However, I do not equate this strong performance with novelty.
> 2. Thank you for correcting my *factual misunderstanding* regarding the number of datasets used in your experiments.
> 3. Regarding the time complexity aspect, thank you for (unwillingly) updating the time complexity according to my comment, but you do not have to. I want to clarify that my intention was not to necessitate a revision but rather to suggest a more accurate representation of the time complexity for the benefit of readers.
> 4. I am pleased to see the additional efforts towards enhancing the reproducibility of your work.
>
> Based on the above points and the information provided in your response, I will maintain my initial rating of the paper and lower my confidence score slightly.

---

> > ### Author Response · Authors · 2023-11-22
> >
> > We really appreciate your constructive feedback!

---

### Official Review · Reviewer_WHhA · 2023-11-17

**Soundness:** 2 fair
**Presentation:** 3 good
**Contribution:** 2 fair
**Rating:** 5
**Confidence:** 4

**Summary:**

This research paper introduces a novel approach to unsupervised information retrieval using large language models (LLMs), focusing on the drawbacks of existing point-wise and list-wise ranking methods when paired with LLMs. The authors propose a pair-wise prompting strategy, featuring two paradigms: scoring mode and generation mode, to address these limitations. They highlight the inefficiency of calibrated scoring in point-wise methods and the inconsistencies in list-wise methods. To enhance efficiency, they employ advanced ranking algorithms and sliding window techniques. This approach simplifies the ranking process by only requiring relative scores between two documents, significantly reducing the need for costly LLM API calls. Extensive testing on public benchmarks reveals that open-source LLMs, even those smaller in scale, can achieve competitive or superior results compared to larger, more complex models like GPT-4. This demonstrates a promising advancement in document ranking using unsupervised LLMs.

**Strengths:**

+ Addresses an important and current task in information retrieval using large language models (LLMs).

+ Conducts many experiments on two public benchmarks, demonstrating the effectiveness of the proposed pair-wise prompting strategies.

+ Achieves competitive results with open-source LLMs compared to larger models like GPT variants, balancing performance and cost-efficiency.

+ Proposes various innovative methods (all-pair, sorting-based, sliding-window) to decrease the number of LLM calls.

**Weaknesses:**

- Limited Novelty: The idea of pair-wise comparison is not entirely new, as it's a common paradigm in the field of information retrieval and has been explored in previous works. In addition, the concept of the sliding window strategy for search acceleration has also been previously discussed in the literature.

- Lack of Code Availability for Reproduction: The absence of shared source code and specific prompts used in the experiments makes it challenging for others to reproduce and verify the results. Moreover, given the reliance on open-source LLMs, there is a concern that the outcomes might significantly vary depending on the choice of LLMs used.

- Inconsistency in Experimental Results: The research shows a significant inconsistency in the performance of the proposed PRP-Sliding-10 method across different benchmarks, specifically TREC-DL2019 and TREC-DL2020. It raises concerns when the method outperforms RankGPT with GPT4 in TREC-DL2020, yet fails to surpass RankGPT in TREC-DL2019. This discrepancy in results between similar benchmarks indicates potential issues with the method's robustness or adaptability across varying datasets. This inconsistency might be due to factors like varying LLM architectures or differing data sets, which can significantly impact the replicability and applicability of the research in real-world scenarios.

- Mismatch with Conference Focus: A significant weakness of the paper is its apparent misalignment with the core themes of the International Conference on Learning Representations (ICLR). Given that ICLR primarily concentrates on advancements in learning representations, the lack of discussion or exploration into this area in the paper stands out. This omission could limit the paper's relevance and appeal to the ICLR audience, who expect contributions that delve into learning representation theories, methodologies, or applications. This paper seems more suitable for ICPE (International Conference on Prompt Engineering).

**Questions:**

Please refer to Weaknesses.

---

> ### Author Response · Authors · 2023-11-20
> **Good luck with your submission to "International Conference on Prompt Engineering"**
>
> The following comment only reflect opinion of the fist author and does not represent the other authors.
>
> Good luck with your submission to "International Conference on Prompt Engineering". Just decline the review invitation next time rather than copying from other reviewers and generating review with LLM.

---

> > ### Author Response · Authors · 2023-11-20
> > **We copy the initial review for posterity**
> >
> > Strengths:
> > Addresses an important and current task in information retrieval using large language models (LLMs).
> >
> > Conducts many experiments on two public benchmarks, demonstrating the effectiveness of the proposed pair-wise prompting strategies.
> >
> > Achieves competitive results with open-source LLMs compared to larger models like GPT variants, balancing performance and cost-efficiency.
> >
> > Proposes various innovative methods (all-pair, sorting-based, sliding-window) to decrease the number of LLM calls.
> >
> > Weaknesses:
> > Limited Novelty: The idea of pair-wise comparison is not entirely new, as it's a common paradigm in the field of information retrieval and has been explored in previous works. In addition, the concept of the sliding window strategy for search acceleration has also been previously discussed in the literature.
> >
> > Lack of Code Availability for Reproduction: The absence of shared source code and specific prompts used in the experiments makes it challenging for others to reproduce and verify the results. Moreover, given the reliance on open-source LLMs, there is a concern that the outcomes might significantly vary depending on the choice of LLMs used.
> >
> > Inconsistency in Experimental Results: The research shows a significant inconsistency in the performance of the proposed PRP-Sliding-10 method across different benchmarks, specifically TREC-DL2019 and TREC-DL2020. It raises concerns when the method outperforms RankGPT with GPT4 in TREC-DL2020, yet fails to surpass RankGPT in TREC-DL2019. This discrepancy in results between similar benchmarks indicates potential issues with the method's robustness or adaptability across varying datasets. This inconsistency might be due to factors like varying LLM architectures or differing data sets, which can significantly impact the replicability and applicability of the research in real-world scenarios.
> >
> > Mismatch with Conference Focus: A significant weakness of the paper is its apparent misalignment with the core themes of the International Conference on Learning Representations (ICLR). Given that ICLR primarily concentrates on advancements in learning representations, the lack of discussion or exploration into this area in the paper stands out. This omission could limit the paper's relevance and appeal to the ICLR audience, who expect contributions that delve into learning representation theories, methodologies, or applications. This paper seems more suitable for ICPE (International Conference on Prompt Engineering).

---

### Official Review · Reviewer_aHw8 · 2023-11-17

**Soundness:** 3 good
**Presentation:** 3 good
**Contribution:** 2 fair
**Rating:** 3
**Confidence:** 4

**Summary:**

The authors propose an unsupervised approach for document ranking using a pair-wise prompting strategy. The primary motivation is to overcome the limitations of the point wise and list wise ranking approaches. The authors conduct experiments on multiple datasets to demonstrate the effectiveness of the PRP approach as compared to both supervised and unsupervised methods. Further, the authors also aim to make the approach more cost-friendly. They have demonstrated the performance of the various optimization approaches as well.

**Strengths:**

The paper proposes a novel solution to ranking using LLMs
The authors have demonstrated their results on several datasets and compared them to several approaches.
The paper also focused on cost efficiency.

**Weaknesses:**

1. The paper seems less suitable for ICLR, but to another venue that focuses on practical/ applied aspects of using LLMs for information retrieval. The approaches are more incremental.

2. The authors have a note in the appendix about reproducibility, but the details are limited. Instead of pointing to Figure 2 as an example, a better description of the prompting approach would be helpful. This is more important given that the authors are aiming to aid academic research, as noted in multiple places across the paper.

3. As for the efficiency, the authors seem to be focusing on relatively smaller set of documents and ranking problems, where they choose 10 out of 100 documents. It will be good to know the performance on larger sets.
Moreover, a quick-sort like partition algorithm might yield better results than a bubble-sort based approach. May be that is why PRP-Sliding-10 has better NDCG@1 but lower NDCG@5 and NDCG@10 than PRP-Sorting.

**Questions:**

Noted in the weakness section.

---

> ### Author Response · Authors · 2023-11-20
> **Thank you for your comment**
>
> We appreciate your comment!
>
> - 1. We respect your opinion while believe it is subjective.
> - 2. We explicitly added the prompt template in the new version.
> - 3. We are confused about this comment. We are ranking 100 documents. AllPair and Sorting clearly do full ranking. Sliding window targets to optimize Top-K metrics, but the ranking of other documents change along the process and they have initial ranking as backup. Thus we believe there's significant misunderstanding of the problem. Also, this is the exact setting of related work. Also, we are confused that studying another sorting algorithm is critical for this work.

---

### Author Response · Authors · 2023-11-15
**General reply**

We appreciate the reviewers’ efforts to comment on the paper. While we wait for the 3rd review, we want to point out some *factual* misunderstandings in terms of the novelty, scope, and setting of the work in the existing reviews.

We acknowledge other constructive comments from the reviewers, in particular,
- We updated the Related Work section
- We added a Reproducibility section at the end of the paper in the updated version.